# Barriers to Water and Sanitation Safety Plans in Rural Areas of South Africa—A Case Study in the Vhembe District, Limpopo Province

**Arinao Murei** [1,*]**, Barbara Mogane** [1]**, Dikeledi Prudence Mothiba** [1]**, Opelo Tlotlo Wryl Mochware** [1]**,
Jeridah Matlhokha Sekgobela** [1]**, Mulalo Mudau** [1]**, Ndamulelo Musumuvhi** [2]**,
Colette Mmapenya Khabo-Mmekoa** [3]**, Resoketswe Charlotte Moropeng** [3]
**and Maggie Ndombo Benkete Momba** [1,*]

1    Department of Environmental, Water and Earth Sciences, Arcadia Campus, Tshwane University of Technology, 175 Nelson Mandela Avenue, Arcadia, Pretoria 0001, South Africa; barbaramogane@gmail.com (B.M.); mothibadeekay@hotmail.com (D.P.M.); mochwareopelo@gmail.com (O.T.W.M.); sekgobelajeridah@outlook.com (J.M.S.); mudaumulalo21@gmail.com (M.M.)
2    Department of Environmental Sciences, University of Venda, Thohoyandou 0950, South Africa; musumuvhi96@gmail.com
3    Department of Biomedical Technology, Arcadia Campus, Tshwane University of Technology, Pretoria 0001, South Africa; mmekoakcm@tut.ac.za (C.M.K.-M.); moropengrc@tut.ac.za (R.C.M.)
*    Correspondence: mureiarinao@gmail.com (A.M.); mombamnb@tut.ac.za (M.N.B.M.)

**Abstract:** The implementation of water and sanitation safety plans (WSSP) has the potential of greatly improving the challenge of resource-limited drinking-water supplies. However, the most effective tool to make WSSP successful is understanding of the factors that contribute to hindering the implementation of these plans, specifically in rural communities. This study therefore aimed at assessing the status of basic services and determining the factors that contribute to hampering the process of WSSP in rural communities. A survey was conducted between March 2020 and March 2021 in rural communities of the Vhembe District, Limpopo Province, South Africa. The overall results indicate that poverty, unemployment, lack of access to purified water, and inadequate sanitation facilities have resulted in waterborne diseases reported within the communities and have a major impact in hindering WSSP. Other barriers observed are inequality regarding financial power, absent and degrading water and sanitation infrastructures, and lack of protection and maintenance of natural water sources. Therefore, there is a need for community members to be educated on proper behavior and perceptions towards sanitation, including working hand-in-hand with different stakeholders, men and women from communities, and different cultures and religions to overcome these barriers, so that human disease associated with water supply, wastewater reuse, and sanitation in rural communities can be alleviated.

**Keywords:** water safety plans; sanitation safety plans; rural communities; water sources; basic services

## 1. Introduction

Access to safe drinking-water sources, appropriate sanitation facilities, and good hygiene are not only fundamental to the health and survival of the people but also to the economic growth and the development of a country. In low- and middle-income countries, inadequate drinking water, sanitation, and hygiene have adverse effects in non-household settings, such as schools, health care facilities, and workplaces. This, in turn, affects the health, education, welfare, and productivity of populations. Inadequate hand hygiene practices have been estimated to affect 80% of the population globally [1].

In December 2019, the coronavirus (COVID-19) pandemic outbreak that emerged in China quickly spread in different parts of the world. South Africa reported the first case of COVID-19 in March 2020. This disease affects the upper respiratory tract, including the nose, throat, and sinuses, and the lower respiratory tract, including the lungs and windpipe. One of the measures to prevent or reduce the spread of COVID-19 suggested by the World Health Organization (WHO) and United Nations Children's Fund (UNICEF) is washing hands frequently and appropriately [2]. However, some communities have trouble accessing water and sanitation resources [3]. Hence, this increases poverty and vulnerability, especially now in the era of COVID-19 [4]. In the face of this pandemic, the South African government organised the emergency supply of water-storage tanks, water trucks, and sanitisers to water-stressed communities [5]. A report by social development in 2020 articulated that during COVID-19 lockdown, the department of water and sanitation delivered a total of 262 water tanks and 27 water tankers in Limpopo. From the 262 water tanks, 132 were delivered in the Vhembe district of the Limpopo province.

The water and sanitation challenges in rural areas of South Africa come as no surprise, as these have been in existence for many decades and continue to plague the poor. A sufficient supply of safe drinking water is a necessity [6,7]. The Limpopo province is recognised as one of the water-stressed South African provinces, with the Vhembe district being strongly impacted by water crisis [8]. It is worth noting that approximately 33% of the South African population lived in rural areas as of 2019 [9], and millions of people are still lacking access to safe drinking water and adequate sanitation. Since South Africa has been recognised as a water-scarce country, there is a need to act promptly to protect its water sources. The available water sources need to be managed in a sustainable and more efficient manner going forward to curb the consequences that go hand in hand with the shortage of water.

Humans are greatly affected by the lack of water within the population, which increases the burden of waterborne diseases and affords them a poor standard of living. It is estimated that over 1.8 billion people worldwide get their drinking water from contaminated water sources [10]. Within developing countries, outbreaks of diseases caused by bacterial and other enteric pathogens remain widespread. People become susceptible to waterborne diarrhoea-causing pathogens such as *Vibrio*, *Cryptosporidium*, and *Salmonella*, to name just a few. In the Vhembe District Municipality (VDM), faecal pollution of water sources has remained on the rise, rendering most of them unfit for human consumption [11]. With a large part of the population dependent on surface water for drinking and other domestic uses [12], the urgency of the situation is evident.

The lack of basic sanitation is a major contributory factor to deteriorating water quality. Adequate sanitation, therefore, has been declared as a fundamental human right by the United Nations [7]. Many people in rural areas lack adequate sanitation, resulting in open defecation, which includes defecating in waterways, open fields, bushes, and forests [13]. This has been a critical aspect of polluting water sources, especially surface water. Conversely, groundwater is mostly contaminated through on-site sanitation amenities in the form of pit latrines and septic tanks. Groundwater is considered a better alternative source of drinking water than surface water, while its quality is not always up to the standard. Many people in low-income countries often use groundwater without initial treatment, which similarly predisposes them to significant health risks [14]. Studies conducted by previous investigators have pointed out that poor groundwater quality in South African rural communities poses a threat to public health [15–17]. Failure of rural schemes to provide potable water to their communities remains a matter of concern [18].

The era of the sustainable development goals of 2030 necessitates an increased effort in ensuring that every individual has their basic needs met irrespective of their demography and social standing [19]. To achieve this, the WHO has developed comprehensive risk assessment and risk management plans for different uses of water, namely water safety plans (WSP), sanitation safety plans (SSP), European Union (EU) bathing water profiles, shellfish sanitary surveys, national plans, and others. The WSP and SSP have been adopted

worldwide and implemented successfully in various countries under different cultural and economic circumstances [20]. However, the flexibility of the tools is unrivalled; they remain unique to a given scenario and limit the appropriation of the same plan under a set of different factors, making it challenging to develop in some low-income settings where pivotal resources are limited.

The WSP considers all steps in a drinking-water supply system, ensuring that water meets regulatory standards and is safe for human consumption [21]. Similarly, the SSP was designed to address the safety of wastewater, greywater, and excreta handling along the entirety of the sanitation service chain [19]. These measures are put in place to minimise as much as possible the human exposure to disease-causing pathogens caused by lack of hygiene. The feasibility, however, is restricted when they are vetted as stand-alone postulates. There is more reason to revitalise the WSP and SSP in order to harmonise the tools already in existence, but they seemingly achieve a defective state when utilised separately. For impoverished rural communities in many African countries, the WSP and SSP implementation remains unachievable.

In 2014, Samwel et al. combined water safety plans and sanitation safety plans into water and sanitation safety plans for risk assessment and management in rural communities; it was successfully applied in the EU since it was site-specific. However, the implementation of this water and sanitation safety plan in the African region to date has been hampered by recurring problems, such as skills shortages within water and wastewater utilities and governmental organisations, competing priorities between utilities and government departments, and poor stakeholder engagement [22]. Although global science continues to provide new insights into effective water and sanitation management, these advances are failing to noticeably improve the livelihoods of needy communities, especially those residing in rural areas. Communities in these areas use available water sources for multiple purposes, including drinking, crop irrigation, livestock rearing, and bathing. To defy the odds, people in the rural area always make the means to access safe drinking water and live in a dignified space with adequate sanitation; there is only so much a layman can do. Authorities, government, and service providers retain the capacity and responsibility to resolve such deeply embedded practices that threaten human lives. Therefore, this current study aimed to identify barriers hindering the success of current risk assessment and risk management plans utilised by local government municipalities of VDM. Furthermore, the findings of this study will be used to compare with the global science innovation to find an effective long-term workable solution for the provision of safe drinking water in impoverished rural communities. Hence, this study acknowledges the recent global achievements of the WSP and the SSP as a sustainable way to reduce risks to health in all human activities associated with the water cycle but equally recognises the barriers that have effectively prevented the application of these approaches in rural Africa.

## 2. Methodology

*Ethical Approval*

This study was conducted taking into consideration the requirement of the ethics clearance approved by the Faculty of Science Research Ethics Committee (FCRE) at the Tshwane University of Technology (TUT), where the study was registered. Access to the Vhembe District Municipality was obtained through the municipal committee upon explaining the purpose of the project. Furthermore, the community members granted permission through their chief or tribal authorities to conduct the study within the targeted areas. Households were randomly selected for participation, and they were given informed consent forms to sign at the beginning of the project. The study expectations and respective obligations by both the participants and investigators were explained, and any question was answered. The participants were not subjected to risks of any kind as a result of this study. Scheme 1 below shows the study design.

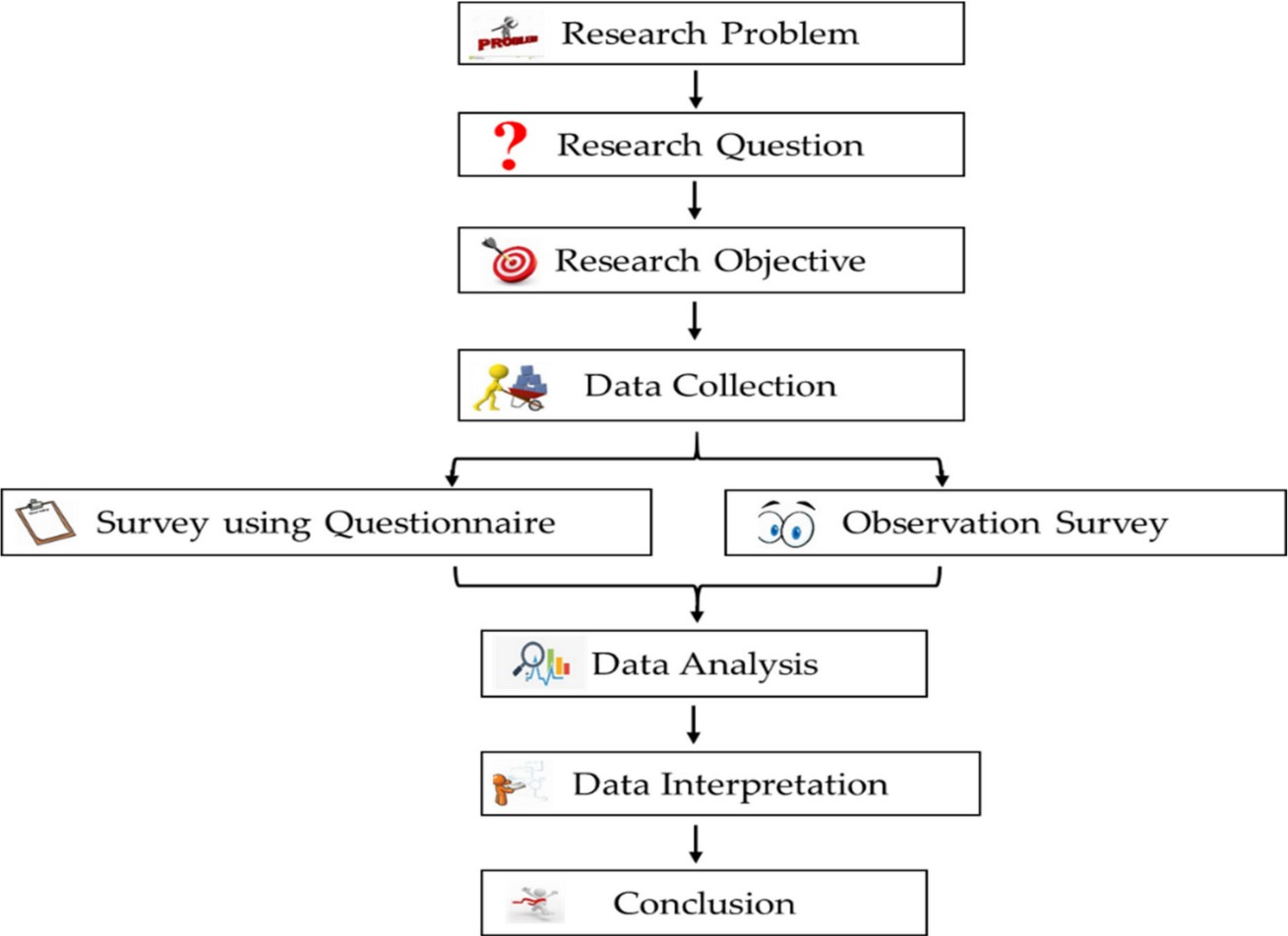

**Scheme 1.** Showing how this work was designed.

## 3. Description of the Study Area

The current study was conducted in the Vhembe District Municipality, which is in the far north of the Limpopo province in South Africa. It constitutes four local municipalities (LM) (Collins-Chabane (CC), Makhado (MK), and Musina and Thulamela (TM)), which are mostly rural, with people of different religious, educational, and socio-economic backgrounds living in communities with distinctly different levels of sanitation. Figure 1 depicts a locality map comprising three selected LM of VDM, the selected villages, water treatment plants (WTP), wastewater treatment plants (WWTP), and the major river streams. Vhembe District municipality has been reported to have water scarcity issues, with the majority of the villages depending solely on contaminated surface-water sources [8,23].

For this study, 35 villages (mostly rural) were randomly selected from three different municipalities. The scarcity of adequate safe drinking water and appropriate sanitation facilities observed during the visits and pictures (Figures 2 and 3) were documented in photos with the consent of the participants.

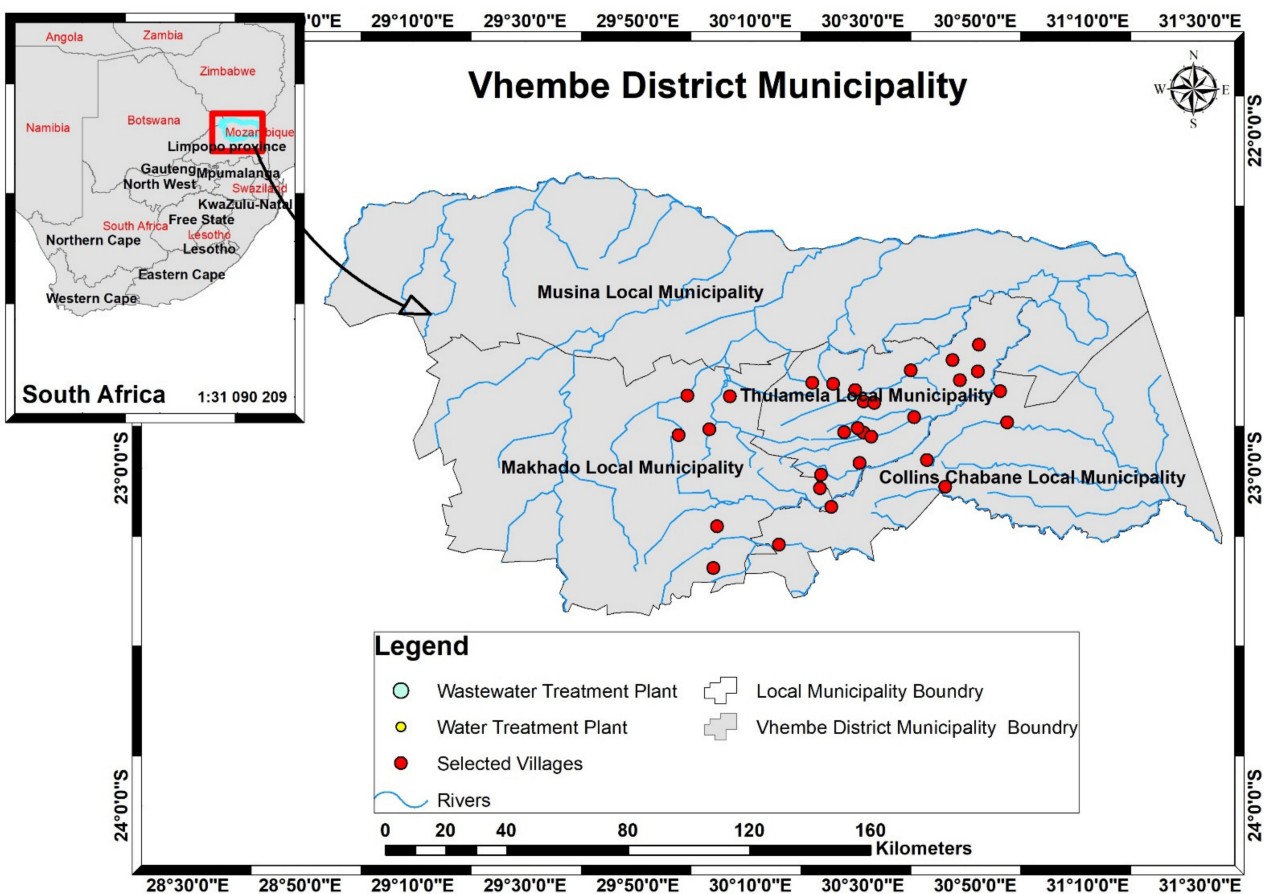

**Figure 1.** A map showing all four LMs of VDM and the selected areas where the study was conducted. Red circles on the map are the representation of the selected study areas from three municipalities.

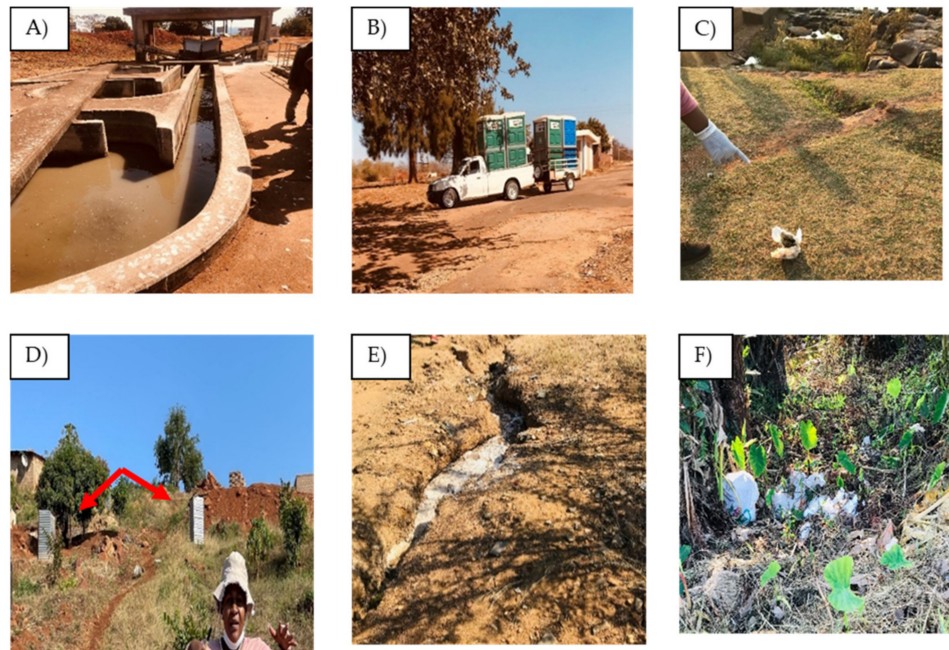

**Figure 2.** Photographs of sanitation facilities and WWTP observed around the study area: (**A**) Tswinga WWTP in Thulamela, (**B**) mobile toilets to be emptied at local WWTP, (**C**) diapers next to Luvuvhu River in Mhinga, (**D**) pit latrines in Tshivhulani households, (**E**) open defecation run offs next to a river in Mhinga, and (**F**) open defecation next to Nandoni Dam.

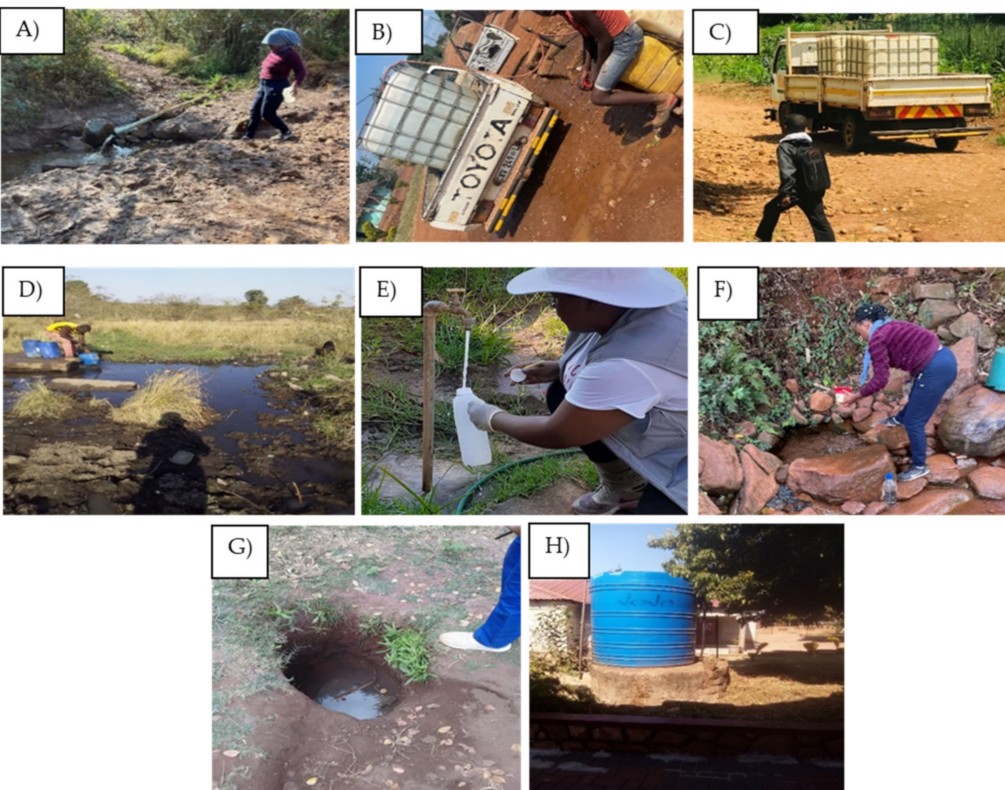

**Figure 3.** Photographs of different water sources (improved and unimproved) observed around the study area. (**A**) Piped spring in Tshivhulani used by community members, (**B**) communal tap in Makuleke, (**C**) water-tanker truck arriving in Dzingahe, (**D**) piped spring in Dididi used by surrounding community members, (**E**) tap in Manini dwelling, (**F**) unprotected spring in Tshilapfene used by community members, (**G**) hand-dug well in Tshivhulani dwelling, and (**H**) borehole water stored in Jojo tank in Tsianda dwelling.

### 4. Data Collection

For this study, a structured questionnaire was divided into two aspects. The first aspect focused on socio-demographic characteristics of household members, which includes the age, education, employment, monthly income, and water-related illness affecting people in households. The second aspect focused on water, sanitation, and hygiene (WASH) characteristics, which includes water sources, household's water treatment methods, sanitation facility data, and hygiene practices. In addition, participants were asked similar questions, such as whether the water supply was working satisfactorily or not; what he major causes of water and sanitation problems in the community were and elaborations on some answers where their perceptions were sought. The questionnaire was translated into local native languages to engage Tshi-Venda- and Xi-Tsonga-speaking participants fully. Data were collected over one year between March 2020 and March 2021 from 1388 households (520 in TM, 442 in CC, and 426 in MK) where one elderly person was selected to respond to the questionnaire. About 5% of the total number of households per village was selected randomly; however, for villages with the highest (≥1000) number of households, a maximum of 50 households was randomly selected per village. The survey was also conducted at WTPs and WWTPs. A total of five WTPs were visited in the VDM. The questionnaire was used to obtain data on the provisioning of drinking water to communities, treatment methods, types of chemicals used, and working conditions. Two sewage treatment plants were visited, and a questionnaire was implemented for each plant. Different anthropogenic activities that influence water quality were observed in the study area.

## 5. Statistical Analysis

All data collected were analysed using statistical software (Stata 14.2). Descriptive statistics was applied for a qualitative analysis. The *t*-test was applied to paired samples to determine whether or not there was a significant difference between the two testing seasons. Significance was considered when the *p*-value was less than 0.05.

## 6. Results

### 6.1. Socio-Demographic Data

During the study period, the population size of the three selected Vhembe district municipalities was 5987. Table 1 shows the main socio-demographic characteristics and water-related illnesses of the study population. The results of the survey revealed that the three selected VDMs are populated by people of age group varying from 26 to 55 years with about 32.9% in TM, 32.7 in CC, and 36.3% in MK. The results revealed that most of the VDM community members had matric as the highest qualification with CC (58.8%), with the highest number of matriculants followed by TM (55%) and MK (51.1%). Of all the surveyed households, 25.9% (MK), 23.4% (TM), and 20.4% (CC) were either at tertiary level or had a tertiary qualification. Moreover, 3.9% (CC), 2.0% (MK), and 1.4% (TM) household members did not attend school.

**Table 1.** Socio-demographic characteristics and history of water-related diseases.

| Characteristics of Household's Respondents | | Local Municipalities | | | | | |
|---|---|---|---|---|---|---|---|
| | | Thulamela | | Collins-Chabane | | Makhado | |
| | | Number | Percentage | Number | Percentage | Number | Percentage |
| Age groups | 0–5 | 263 | 12.5% | 320 | 16.0% | 230 | 12.2% |
| | 6–15 | 386 | 18.4% | 396 | 19.8% | 358 | 19.0% |
| | 16–25 | 400 | 19.1% | 350 | 17.5% | 345 | 18.3% |
| | 26–55 | 691 | 32.9% | 653 | 32.7% | 686 | 36.3% |
| | ≥56 | 360 | 17.1% | 280 | 14.0% | 269 | 14.3% |
| | | *n* = 2100 | | *n* = 1999 | | *n* = 1888 | |
| Educational level | No schooling | 15 | 1.6% | 31 | 3.9% | 20 | 2.0% |
| | Primary | 88 | 8.0% | 48 | 6.0% | 64 | 6.5% |
| | Secondary | 135 | 12.2% | 87 | 10.9% | 141 | 14.4% |
| | Matric | 607 | 55.0% | 469 | 58.8% | 501 | 51.1% |
| | Tertiary | 258 | 23.4% | 163 | 20.4% | 254 | 25.9% |
| | | *n* = 1103 | | *n* = 798 | | *n* = 980 | |
| Number of employed individual(s) in the household | 0 | 228 | 44.1% | 188 | 45.0% | 152 | 35.9% |
| | 1 | 190 | 36.7% | 134 | 32.1% | 167 | 39.4% |
| | 2 | 82 | 15.9% | 79 | 18.9% | 78 | 18.2% |
| | 3 | 16 | 3.1% | 14 | 3.4% | 24 | 5.7% |
| | ≥4 | 1 | 0.2% | 3 | 0.7% | 3 | 0.7% |
| | | *n* = 517 | | *n* = 418 | | *n* = 424 | |
| Monthly salary distribution | ≤ZAR 5000 | 1149 | 91.6% | 240 | 43.0% | 199 | 6.7% |
| | ZAR 5000–ZAR 10,000 | 83 | 6.6% | 303 | 54.3% | 73 | 24.6% |
| | ≥ZAR 10,000 | 23 | 1.8% | 15 | 2.7% | 25 | 8.4% |
| | | *n* = 1255 | | *n* = 588 | | *n* = 279 | |
| Illnesses affecting VDM population | Diarrhoea | 93 | 17.9% | 71 | 16.1% | 45 | 10.6% |
| | Trachoma | 8 | 1.5% | 7 | 1.6% | 9 | 2.1% |
| | Body lice | 3 | 0.6% | 0 | 0% | 0 | 0% |
| | Rash | 21 | 4.0% | 21 | 4.8% | 2 | 0.5% |
| | None | 395 | 76.0% | 343 | 77.6% | 368 | 86.4% |
| | | *n* = 520 | | *n* = 442 | | *n* = 426 | |
| Does diarrhoeal frequency occur | Yes | 18 | 12.3% | 430 | 7.6% | 418 | 7.8% |
| | No | 502 | 87.7% | 430 | 92.4% | 418 | 92.2% |
| | | *n* = 520 | | *n* = 442 | | *n* = 426 | |

Collins-Chabane had the highest unemployment rate of 45.0%. Overall, the survey showed that the majority of the MK community members are employed; a high number of people in VDM are earning less than ZAR 5000 per month. In all LMs, the number of respondents earning salaries at the range of ZAR 10,000–ZAR 15,000 was very low. The findings showed that many people have good health since there was a high percentage in all three municipalities showing no illness. Nevertheless, diarrhoea was observed to be more common in all municipalities, with 17.9%, 16.1%, and 10.6% in TM, CC, and MK LMs, respectively. Further, more than 7% in all municipalities reported to have frequent diarrhoea. Rash and trachoma were also observed, and only a few respondents in TM indicated that they do suffer from body lice, especially young children.

*6.2. Water, Sanitation, and Hygiene (WASH) Data in Vhembe District Municipality*

Table 2 shows WASH characteristics in rural communities of the VDM, where residents rely on different improved and unimproved water sources. The findings showed that the majority of the VDM obtain water from communal taps (for public use) and standpipes within the yards; most households of CC had taps installed within the yards (58.6%), followed by MK (56.2%) and TM (48.5%). Just over 2% of TM households collects water directly from the river, while 1.7% of the households depends on rain-harvested water. An overwhelming majority of the VDM showed to have no knowledge of the different household water treatment methods. The communities' knowledge of household water treatment methods is very poor: 24% in MK, 27% in TM, and 27% in CC (Table 2).

The most used sanitation facilities include the pit latrine and pit with concrete slab. About 3.6% of households in CC indicated that they still practice open defecation, while 1.1% of the households rely on their neighbour's pit latrine. Very few households have a ventilation-improved pit latrine (VIP). About 34.9% TM, 39.8% CC, and 31.2% MK of households contain sanitation facilities built by the municipality and about 34.5% TM, 24.7% CC, and 18.5% MK have sanitation facilities built by the household members; a total of 26.2% TM, 27.6% CC, and 47.8% MK of the households have hired personnel, while a total of 4.2% TM, 7.9% CC, and 92.4% MK of the households have no toilets. Those households without sanitation facilities might share the facilities with neighbours or practice open defecation.

Among 123 households that used a sanitation facility with a hand-wash basin, 9.2% were found in TM, 14.9% in MK, and 5.7% in CC. It is also important to mention that there was no hand-wash basin in a high number of households: (90.8% in TM, 87.1% in MK, and 94.9% in CC). On the question regarding the respondents' opinions on whether they wash their hands with soap or not, results revealed that most people did wash their hands with soap. Results also revealed that some of the households spent their income for the purchase of household cleaning products. However, about 60.05% in TM, 49.6% in CC, and 55.7% in MK do not spend on such reagents.

*6.3. Water Treatment Plants*

6.3.1. Potable Water

The results indicated that the WTW1 up to WTW5 employ different disinfection methods. As can be seen in Table 3, the disinfection method used in all drinking-water treatment is chlorination. However, there are many types of chlorine: chlorine gas, sodium hypochlorite (liquid), lithium hypochlorite (granular), calcium hypochlorite (granular), dichlor (tablets/granular), and trichlor (tablets/granular). The WTW2 and WTW3 reported that they used gas chlorine, and WTW1 used granules (calcium hypochlorite) because their gas glass is broken, while WTW4 and WTW5 reported using both the gas and granules (calcium hypochlorite). Chlorine is always available in all treatment plants; however, WTW1 indicated that they have difficulties in calculating the chlorine dosage.

While the WTW2, -3, and -4 measure both pH and turbidity every two hours, WTW1 and -5 only measure pH every two hours. The WTWs do not have an on-site laboratory for microbial tests; however, samples are sent to the VDM laboratory once a week or once every

two weeks to test faecal contamination and the pathogens. It was noted that WTW1 and WTW5 equipment need replacement, whereas other plants' equipment is up to standard. Furthermore, WTW1 and WTW5 reported not supplying enough water to the community because of broken infrastructure such as blocked pipelines, non-functioning water pump, high water demand, and WTW2, -3, and -4 supplying enough water.

**Table 2.** Characteristics of WASH distribution across the three LMs in VDM.

| Characteristics of Household's Respondents | | Local Municipalities | | | | | |
|---|---|---|---|---|---|---|---|
| | | Thulamela | | Collins-Chabane | | Makhado | |
| | | Number | Percentage | Number | Percentage | Number | Percentage |
| Main water sources | Communal tap * | 223 | 42.8% | 178 | 40.2% | 128 | 30.1% |
| | Tap in dwelling * | 252 | 48.4% | 252 | 57.1% | 239 | 56.2% |
| | Rain | 9 | 1.7% | 0 | 0% | 0 | 0% |
| | Borehole | 21 | 4.1% | 0 | 0% | 48 | 11.3% |
| | Spring | 4 | 0.8% | 3 | 0.7% | 10 | 2.4% |
| | Truck-tanker * | 0 | 0% | 0 | 0% | 0 | 0% |
| | River | 11 | 2.1% | 9 | 2.0% | 1 | 0.2% |
| | Bottled * | 0 | 0% | 0 | 0% | 0 | 0% |
| | | | *n* = 520 | | *n* = 442 | | *n* = 426 |
| Knowledge of household water treatment methods | Do not know | 193 | 37.1% | 191 | 45.3% | 188 | 44.3% |
| | Bleaching/Chlorination | 105 | 20.2% | 103 | 24.5% | 105 | 24.8% |
| | Boiling | 153 | 29.4% | 100 | 23.8% | 106 | 24.8% |
| | Salting | 13 | 2.5% | 4 | 1.0% | 1 | 0.2% |
| | Bleaching/Boiling/Salting | 11 | 8.7% | 1 | 5.2% | 0 | 0.0% |
| | Other | 45 | 2.1% | 45 | 0.2 | 25 | 5.9 |
| | | | *n* = 520 | | *n* = 421 | | *n* = 425 |
| Water treatment methods usage in households | Do not use | 375 | 72.2% | 374 | 72.1% | 393 | 75.6% |
| | In use | 144 | 27.8% | 145 | 27.9% | 126 | 24.4% |
| | | | *n* = 520 | | *n* = 442 | | *n* = 426 |
| Sanitation facilities used | Pit | 366 | 70.4% | 268 | 60.6% | 288 | 67.6% |
| | Pit with concrete slab | 122 | 23.5% | 127 | 28.7% | 91 | 21.4% |
| | Flush septic | 16 | 3.1% | 1 | 0.0% | 36 | 8.5% |
| | VIP | 1 | 0.0% | 1 | 0.0% | 1 | 0.0% |
| | Open defecation | 7 | 1.3% | 16 | 3.6% | 9 | 2.1% |
| | Neighbours | 7 | 1.3% | 5 | 1.1% | 0 | 0% |
| | | | *n* = 520 | | *n* = 442 | | *n* = 426 |
| Who built sanitation facility | Municipality | 181 | 34.9% | 176 | 39.8% | 116 | 31.2% |
| | household member | 179 | 34.5% | 109 | 24.7% | 69 | 18.5% |
| | Hired personnel | 139 | 26.2% | 122 | 27.6% | 178 | 47.8% |
| | None/no toilet | 23 | 4.2% | 35 | 7.9% | 9 | 2.4% |
| | | | *n* = 519 | | *n* = 442 | | *n* = 372 |
| Is the sanitation facility still in good condition | Yes | 469 | 90% | 380 | 86% | 363 | 85% |
| | No | 50 | 10% | 62 | 14% | 63 | 15% |
| | | | *n* = 519 | | *n* = 442 | | *n* = 426 |
| Availability of hand-washing basin | Yes | 47 | 9.2% | 54 | 14.9% | 22 | 5.7% |
| | No | 464 | 90.8% | 363 | 87.1% | 412 | 94.9% |
| | | | *n* = 511 | | *n* = 417 | | *n* = 434 |
| Proper hygiene practice in study population | Yes | 431 | 82.9% | 403 | 91.2% | 381 | 89.4% |
| | No | 71 | 13.7% | 28 | 6.3% | 36 | 8.5% |
| | Sometimes | 18 | 3.5% | 11 | 2.5% | 9 | 2.1% |
| | | | *n* = 520 | | *n* = 442 | | *n* = 426 |
| Ability to purchase cleaning product | Yes | 161 | 39.9% | 128 | 50.4% | 172 | 44.3% |
| | No | 242 | 60.05% | 126 | 49.6 | 216 | 55.7% |
| | | | *n* = 403 | | *n* = 254 | | *n* = 388 |

Note: * Treated water.

**Table 3.** Water treatment plant data.

| Water Treatment Works | Type of Chlorine | Availability of the Chlorine | How Often Is Water Tested for Faecal Contaminants and Possible Pathogens? | Does the Plant Provide Enough Water to the Community? | If No, Why? |
|---|---|---|---|---|---|
| WTW1 | Calcium hypochlorite | Always available | Once a year | No | Pipeline blocked |
| WTW2 | Chlorine gas | Always available | Once a week | Yes | NA |
| WTW3 | Chlorine gas | Available for a period of 1 to 2 weeks | Everyday | Yes | NA |
| WTW4 | Calcium hypochlorite and chlorine gas | Always available | Everyday | Yes | NA |
| WTW5 | Calcium hypochlorite and chlorine gas | Always available | Once a month | No | High water demand |

### 6.3.2. Wastewater

Table 4 illustrates results obtained from the wastewater treatment works (WWTW). Both WWTW surveyed indicated that they receive wastewater influents from malls, townships, hospitals, and mortuaries, and the plants can manage the received wastewater influents. Chlorination is the method used to treat the wastewater influents in both plants, and they both indicated that they do not have any difficulties in calculating the dosage required for treatment except during floods. To overcome this, WWTW1 has developed the mechanism of bypassing the wastewater to the maturation ponds, while WWTW2 was not able to overcome this challenge. Following wastewater treatment, the treated effluent is discharged to the rivers. Both these WWTWs were requesting new pumps and mentioned that the sludge treating tanks were not functioning. They emphasised a need for equipment repairs for a better operation.

**Table 4.** Wastewater treatment plant data.

| Wastewater Treatment Plants | Where Is the Wastewater Coming from? | Ability to Effectively Manage the Waste Received | Have You Ever Experienced Any Difficulties during Wastewater Treatment? | If Yes, How Did You Overcome That? | Is the Equipment up to Standard or Need Repair or Replacement? | What Is the Wastewater Used for after Treatment? |
|---|---|---|---|---|---|---|
| WWTW1 | Malls, townships, hospitals, and mortuaries | Able | Yes, during floods | Bypass to maturation ponds | Repairs needed | Supplies water to water sources (rivers) |
| WWTW2 | Malls, townships, hospitals, and mortuaries | Able | Yes, during floods | Nothing | Repairs needed | Supplies water to water sources (rivers) |

### 6.3.3. Main Activities Observed around Water Sources

Different anthropogenic activities were observed around water sources in the VDM. Some of the main activities observed near rivers, dams, and springs include washing

clothes and cars, bathing, animal grazing, swimming, open defecation (from both animals and human), and agricultural activities (especially crops and plant production). Most of the activities were observed within TM LM. In CC LM, however, activities such as laundry, bathing, and the release of treated wastewater was not observed. Finally, in MK LM, there were more activities that were not observed; these were bathing, swimming, fishing, abstraction of water, release of wastewater, and brickmaking.

## 7. Discussion

It is well-known that water and sanitation safety plans can be one way to obtain and maintain safe drinking water and sanitation systems and to minimise related diseases. For WSSP to be successful in rural communities, the water supply and sanitation system needs to be understood, and this also includes the identification of potential hazards that contribute to contaminating water resources. Furthermore, the control measures need to be implemented involving different stakeholders for water and sanitation management [22]. Development of a successful WSSP is not only the responsibility of the water providers or water institutions, but the public needs also to be involved. However, in many rural areas, especially in sub-Saharan Africa, little is known about this important concept. This study aimed at identifying the barriers that hinder the success of water and sanitation safety plans in the rural areas of VDM. For this purpose, a survey was conducted in three selected local municipalities of the VDM. Analysis of the demographic information, water systems, sanitation, and hygiene and health were done to identify local barriers that hinder the success of water and sanitation safety plans in this district municipality.

Several risks to public health were identified as being necessary for developing a water safety plan. These included the plentiful activities observed around unprotected water sources—from both humans and animals; the use of unimproved sanitation facilities and water sources; the release of wastewater into natural water bodies; and the dumping of litter, including diapers, near water ways, to name a few. In households where pit latrines were available, it was observed that they were situated near water sources, especially boreholes. This clearly showed the lack of water and sanitation management. The management of water and sanitation plays an important role in minimising water contamination. Hence, Municipal Water and Sanitation Services Authorities and community leaders should be engaged as stakeholders in WSP and SSP awareness and guidance campaigns. There is a need for transparency in the understanding and acceptance of the planned actions that can lead to community mobilisation to tackle the barriers hindering the management of water and sanitation in rural areas.

The survey revealed that VDM is inhabited by people of age groups varying between 26 and 55 years, findings that are similar to those recorded by South African Census 2016 [24]. It was found that in some households, none of the residents had a tertiary qualification, and roughly 36% did not have a matric certificate, followed by 29% of responses having only one person in the household with a matric certificate. The dynamics of poverty have a major role to play when it comes to the proper implementation of water and sanitation safety plans. The overall frequency of families who have no employed member is 35.9% in MK, which is very low compared with the other LMs. In this study, people with more income reside in CC and MK compared to TM, which had the highest population with the lowest income. The MK and CC LMs could be regarded as having more financial power based on the average overall income that each household receives.

In developing countries, people in rural communities cannot afford to buy bleach or salt for water treatment [25,26]. This was the case in a study done by Moropeng and Momba in the Makwani village where the majority of unemployed people could not afford to purchase the liquid bleach required for the treatment of drinking water in their homes [27]. Similarly, this was observed in the rural communities of the VDM. The major impact of having low income in water and sanitation management is that people drink contaminated water, and as a result, they may experience diarrhoeal disease, whereas those who have higher income can attain proper sanitation and make the means for water

treatment in order to minimise diarrheal diseases. Diarrhoea is known to be more prevalent in immunocompromised individuals; hence, in this study, young children ($\leq$5 years old) were found to be mostly affected as compared to older people ($\geq$65 years old).

Although the VDM has a large number (39) of water supply schemes, some households still remain unserved [25,28]. This study proved that the majority of the VDM households do have standpipes that could improve water supply within the study population. However, with the incidences of water shortages, which can last for weeks, months, or even years, community members tend to rely on any available water source. This erratic water supply increases the dependence on dams, springs, rivers, and small streams for daily drinking-water sources. Our findings show that some of the water sources used at households are unimproved; i.e., in Thulamela, about 5% of the communities use unimproved surface water, which is unprotected and therefore remains open to different anthropogenic activities and the subsequent contamination by faecal matter. Few villages in VDM reported that illegal connections were one of the challenges affecting the water supply in the areas. The underlying causes include inconsistency of water distribution to various sections due to broken WTPs and illegal connections of water supply systems, amongst others. According to the Vhembe Integrated Development Plan (IDP), all households in the VDM are said to be within a water scheme although some still do not have access to a reliable water supply [23]. At some of the communal standpipes, it was observed that there was no water. There is a need for organising a water and sanitation safety plan team that represents the different stakeholders, men and women, different cultures, and religions. This team would be able to set up various tasks and responsibilities and define activities that will be taken into consideration to tackle the challenges facing the community.

According to Mnisi [29], community members must be involved in water projects and decision making. There seemed to be a gap in the communication pipeline between the VDM and the community in decision making and other processes needing engagement; involving community members help with improving water and sanitation service delivery. The result is poor water and sanitation service delivery. For example, they have no choice in selecting the appropriate technology for their community, such as boreholes or hand pumps. Hence, only those who have adequate income can drill their private boreholes, and the poor will continue to be deprived, showing obvious inequality in service delivery. The findings of Khabo-Mmekoa and Momba reported the inequity in the provision of clean and safe drinking water that exists even today in South Africa. They found that in metropolitan areas, the infrastructure for water treatment and distribution to consumers was of higher quality in contrast to rural areas, where it was poor or non-existent [30]. Further, this is most evident in rural communities where only the people with adequate income have private boreholes, while the disadvantaged majority depends on natural sources for drinking water that are unsafe and exposed to contamination.

Although the COVID-19 pandemic rapidly caused devastating socio-economic impacts, such as income loss, business impacts, and health concerns worldwide, it has served as a reminder to water authorities and the government about the importance of the provision of adequate, clean water and proper sanitation to all in developing countries. In the study, in targeted areas, for example, certain volumes of water were sold at ridiculous prices (USD 0.14 for 25 L), or access was obtained by subscription (USD 7.92 per month). By South African law, the minimum quantity of potable water of 25 L per person per day or 6 kL per household per month is a basic standard [31]. This goes hand in hand with ensuring a consumer never goes seven days in a year without supply. This, unfortunately, is still not a reality for many village dwellers. Some have gone as long as 10 years with dry taps. They believed this to result from insufficient government funds or capacity to complete construction projects or service old infrastructure. In the face of COVID-19, the South African government emergency responses have effectively expanded national access to WASH facilities, services, and awareness programmes and platforms. They also organised the emergency supply of water-storage tanks, water tankers, soaps, and alcohol-based hand sanitisers as well as ablution facilities to water-stressed communities [5,32]. However,

not all rural areas received such services. The lack of services creates resistance from end-users, making public participation and engagement with authorities on core issues nearly impossible.

During the survey, some participants pointed out that they were not aware that water should be purified prior to consumption. Others reported that they do not know if the water supplied to them by the municipalities is treated or not. Nevertheless, they mentioned that they have been using the water from other sources, such as dams, rivers, and springs, without any prior treatment, and none had shown any signs of sickness, especially from spring water. Consequently, they draw conclusions that the water is harmless; hence, there is no need to purify it. This may explain the lacking commitment of Water and Sanitation Services Authorities to invest in water and wastewater management.

The results also revealed that many people in all selected LM of the VDM have no knowledge on the different household water treatment methods. There is a significant number of literate people in the three LMs, a majority of whom have acquired a matric certificate. It may be worthwhile to consider avenues to break instilled behaviour and habit when it comes to water purification at household level. This requires the involvement of all stakeholders in the education of community regarding the importance of water quality and sanitation management for public health protection. This means that awareness and information sharing amongst community members needs to be accomplished in a more innovative and easily comprehensible manner. Education may be ruled out as a hindering barrier towards the success of water and sanitation safety plans in the studied population. Our findings are in line with those reported by Angoua and colleagues, as they did not identify the level of education as a risk factor for lacking access to drinking water [33].

Availability of sanitation facilities is necessary to reduce possible faecal–oral transmission through contaminated water by improper faecal matter disposal [34]. Yet, it still remains a major challenge in some rural communities of VDM. One such community in CC has several residents still actively practicing open defecation owing to the uneven distribution of sanitation infrastructure, especially pit latrines. A large percentage of households from the study population uses on-site sanitation facilities such as pit latrines and septic tank system. About 90%, 86%, and 85% in TM, MK, and CC respectively, were observed to be in good condition. However, some mentioned experiencing problems during heavy rain since wastes are lifted, and pit latrines look full, but in fact they are not, driving them to look for other avenues to relieve themselves. This highlights that groundwater mixes with waste, and treatment for groundwater might be important as well as the need for access to improved sanitation facilities. Of those with a sanitation facility, 1239 households used a sanitation facility without a hand-wash basin. This is because most households use pit latrines, which are dry toilets that function without water and hence do not have built-in hand-washing basins. This is of particular concern because the availability and/or visibility of a hand-washing basin affects the perception of hand hygiene. Cloutman-Green and co-workers (2014) [35] reported that the use of handwash basins is proportional to their visibility; people are more likely to wash hands more frequently when sinks are more apparent. Our findings revealed that most people do actually wash their hands more frequently. This impressive improvement in hand washing may be due to the COVID-19 pandemic, which requires people to frequently wash their hands as a preventative measure from infection.

Being of sound financial standing might be grounds for improved sanitation practices. It allows for a budget that fits cleaning reagents, building better sanitation facilities and accessibility to purified water from aloof distances, and using own or rented transportation. The MK LM showed a good trend in the overall perspectives of the study even though communities do not have much affluence compared to CC. When it comes to the usage of open surface-water sources, MK ranks the lowest. Instead, there was an increased usage of underground water, which has less risks of being contaminated compared to surface water. Looking at hygiene practices towards sanitation facilities, the results showed that people

in MK spend more on cleaning reagents. This shows that they are aware that sanitation facilities must be clean for the sake of health.

The results obtained from the WTWs indicated that the functionality of a treatment plant affects the ability of a plant to provide water to the community. Some WWTP infrastructures are not to par and can therefore not cater for wastewater received by the plant. This was evident in reports by WTWs 1 and 5, which indicated that their pieces of equipment are not up to standard and need some replacements. As a result of this, these plants are not able to provide enough water to the communities, resulting in water supply interruptions. Similar results were observed in a study by Mwelase [36], which pointed out that water supply interruptions were due to outdated infrastructures. Hence, it is evident that that there is lack of water and sanitation infrastructure maintenance regularly. Furthermore, inconsistency of microbial water-quality tests in VDM was observed. It was found that one of five WTPs relies on a laboratory which is located far away from the plant, for microbiological testing. This may affect the test, as some samples need to travel for long distances to reach the microbiological laboratory. The WWTWs data showed that staff members have difficulties in measuring chlorine dosage especially during floods. Hence, they indicated that solution is needed for treatment during floods. Both plants WWTW1 and WWTW2 indicated that the wastewater is mostly received from township, shopping centers, and mortuaries. Hence, the municipality should ensure proper management of the domestic waste especially in the rural areas.

The water and sanitation safety plans could be effectively implemented if there is a wide view of all the entities that play a role in the whole process. This investigation explored several aspects that might significantly impact the smooth implementation of the water and sanitation safety plans. The study showed that areas such as TM and CC present issues that could hinder the smooth running of the water and sanitation safety plans. These comprise poverty and unemployment and lack of access to infrastructures to purify water as well as access to adequate sanitation facilities.

## 8. Conclusions

The information that was obtained from the study are juxtaposed with findings from other studies and conspicuously expose the challenges facing the rural areas of the Vhembe district in terms of quality of water and sanitation management. It is evident that there will be many difficulties to implementing proper water and sanitation safety plans in the rural area of the Vhembe District municipality, such as differences or inequality in financial power, absent and degrading water and sanitation infrastructure, lack of protection, and maintenance of natural water sources. With such issues, it will be extremely cumbersome to even start the water safety plan and let alone the sanitation safety plans in an area where there is one major WWTP, which the majority of the populations do not utilise, instead reverting back to pit latrines, which in many cases are not even regulated. However, there is a great possibility that the situation in the rural area can change given that a majority of the population do wash their hands with soap and have shown great interest in being a part of the study from the very beginning; the VDM and many gate keepers were keen to provide better water and sanitation conditions in the lives of their communities and allow for smooth-running water and sanitation safety plans since it follows the water supply chain. There must be great care taken in water sources or catchment areas and avoid activities such as fishing, bathing/swimming, and personal extraction for agricultural purposes, especially where the community draws their water for consumption purpose. Hence, these activities need to be better regulated, and water sources should be protected. The sanitation safety plan, though it is somewhat intricate to deal with because it is more concerned with behaviour and habit, can still change if people in the rural area obtain more insight and education about proper behaviour and perceptions towards sanitation. It may be profitable to start by ensuring people's basic rights are met. More funds need to be allocated by the state for the running of the WSSP. Those funds would need to be strictly monitored to

ensure appropriate usage. It is also important for the people to make use of the resources at hand rather than focusing on dreams. Further awareness is still necessary.

**Author Contributions:** M.N.B.M. conceived the project; M.N.B.M., A.M., D.P.M., B.M., O.T.W.M., J.M.S., M.M. and C.M.K.-M. conceived and designed the experiments; M.N.B.M., A.M., D.P.M., B.M., O.T.W.M., J.M.S., M.M., N.M., C.M.K.-M. and R.C.M. performed the experiments and analysed data. All authors have read and agreed to the published version of the manuscript.

**Funding:** This study was supported by the South African Research Chairs Initiatives (SARChI) in Water Quality and Wastewater Management, funded by the Department of Science and Technology, as administrated by the National Research Foundation (UID87310). Additional funding was received from Tshwane University of Technology.

**Institutional Review Board Statement:** The study was conducted in accordance with the Declaration of Helsinki, and approved by the Tshwane University of Technology Research Ethics committee (FCRE 2019/08/003 (FCPS 03) (SCI) and 20 March 2020).

**Informed Consent Statement:** Written informed consent was obtained from all subjects involved in the study.

**Conflicts of Interest:** The authors declare no conflict of interest. Opinions expressed and conclusions drawn are those of the authors.

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
