# Peer review of "Barriers to Water and Sanitation Safety Plans in Rural Areas of South Africa—A Case Study in the Vhembe District, Limpopo Province"

_water, doi:10.3390/w14081244_

Round 1
Reviewer 1 Report
Title: Barriers to Water and Sanitation Safety Plans in Rural Areas of South Africa–A Case Study in the Vhembe District, Limpopo Province
This work is interesting, and provide some helpful information about the Water Safety in Rural Areas of South Africa. The manuscript is well prepared. Some revisions should be considered before acceptance.
The labels in figure 2 are not well shown.
A schematic image shown the design of this work is helpful.
Some description in the manuscript should be refined to improve the quality.
The full expression of abbreviation should be given when they are firstly mentioned in the manuscript.
Grammar errors and typewriting errors should be fully checked and corrected in the manuscript.
Author Response
Thank you so much for the commets and recommendations. We really appropriate you effort.

Reviewer 2 Report
The paper is well written and there are a very few typographical/English language issues which I have marked with sticky notes.
The data is interesting and useful.
Figures 2 and 3 need improvement – on 2, the letters don’t show. 3b needs to be straightened. All are too small and look as if they have been stretched width-ways.
Table 1 the header needs clarified – it is number as well as frequency and nowhere does it currently explain that the terms Thulamela LM etc are the names of the municipalities. The table should be able to be read alone without reference to the text. The term “employment level” needs to be explained in the table (for example, the education level is clear).
Table 2 don’t use HH – there is plenty of room to write Households and the footnote definition is inelegant.
In both tables, justifying the numbers so there were clear sub-columns of number and % with each one right-justified to be in a neat arrangement would make them easier to read.
Line 259-261 odd phrasing – 10% answered no while 80-90% answered yes. The result is that most people DID wash their hands so don’t make it sound like a bad thing!
Table 5 could probably be replaced with a short description showing the activities NOT found in the few places relevant – in most places they did everything.
Table 6 is very unclear – the header and the footnote about P are not sufficient to explain the content. You also reference things being coloured red, but nothing is coloured at all – do you mean bold face. I think the table shows a positive correlation between matric and illness in TM, matric and illness in MK and Tertiary and illness in TM – but since there are other municipalities where this isn’t so it surely does not imply a causal link? It looks random to me. I think it is unreasonable to claim any association based on this and you should not do so. My recommendation is actually to take out all reference to the Pearson coefficient and make a more qualitative interpretation instead, writing this in the discussion (see below).
The discussion section is very long and does not really lead to a clear conclusion. I think what you are intending to say is that there are significant issues in these localities, and the provision of education, some treatment of water and waste water and other factors does not in itself help all that much. There would therefore be challenges creating a WSSP. You can do this much more concisely than you do.
I recommend that either at the start of the discussion, or earlier in the paper in a background section, you give a clearer summary of what might be needed for a successful WSSP (referencing the appropriate literature in more detail) and then you can relate your findings to whether or not you have what is needed in these municipalities and use your excellent data to recommend which are the most important things to tackle first.
I would like to see this published, mainly because the data is very good and useful for the community of researchers, but the statistics and discussion need significant revision to improve it.

Author Response
Thank you so much for your comments and recommendations. We appreciate your effort.
